

# Seroprevalence of hepatitis B virus in Taiwan 30 years after the commencement of the national vaccination program

Yang-Cheng Hu[1,*], Chih-Ching Yeh[1,2,*], Ruey-Yu Chen[1], Chien-Tien Su[1,3], Wen-Chang Wang[4], Chyi-Huey Bai[1,5], Chi-Fei Chan[6] and Fu Hsiung Su[5,7,8,9]

[1] School of Public Health, College of Public Health, Taipei Medical University, Taipei City, Taiwan
[2] Department of Public Health, China Medical University, Taichung City, Taiwan
[3] Department of Family Medicine, Taipei Medical University Hospital, Taipei City, Taiwan
[4] The Ph.D. Program for Translational Medicine, College of Medical Science and Technology, Taipei Medical University, Taipei City, Taiwan
[5] Department of Public Health, School of Medicine, College of Medicine, Taipei Medical University, Taipei City, Taiwan
[6] Fu Jen Clinic, College of Medicine, Fu Jen Catholic University, New Taipei City, Taiwan
[7] School of Medicine, College of Medicine, Fu Jen Catholic University, New Taipei City, Taiwan
[8] Division of Family Medicine, Department of Community Medicine and Long Term Care, Fu Jen Catholic University Hospital, New Taipei City, Taiwan
[9] Department of Family Medicine, Wan Fang Hospital, Taipei Medical University, Taipei City, Taiwan
* These authors contributed equally to this work.

Corresponding author
Fu Hsiung Su,
williamsufh1@yahoo.com.tw

## ABSTRACT

**Background**. In this study, the long-term efficacy of hepatitis B virus (HBV) vaccination was assessed using seroprevalence and an age–period–cohort (APC) model of HBV seromarkers among university entrants 30 years after the introduction of the national neonatal HBV vaccination program in Taiwan.

**Methods**. In total, data of 17,611 university entrants who underwent university entrance health examinations between 2005 and 2016 were included. The seroprevalence of the HBV surface antigen (HBsAg) and the levels of the antibody against the HBV surface antigen (anti-HBs) in each year group and sex were calculated. The levels of the antibody against the HBV core antigen were examined only for 2012 and 2016. The APC model was used to analyze the HBV carrier rates.

**Results**. The chronic HBV infection (HBsAg positivity) rate decreased from 9.7% in university students born before June 1974 to <1.0% in students born after 1992. The prevalence of anti-HBs positivity declined, particularly between the 1984–1988 cohort (78.2%–53.2%) and the 1990–1994 cohort (60.6%–44.4%). Our APC model revealed that the chronic HBV carrier rate among the student population was affected significantly by age, period, and cohort ($P < 0.001$).

**Conclusions**. HBV vaccination is one of the most effective strategies for preventing HBV infection. However, for complete eradication of HBV infection, the development of strategies that detect vaccination failure more effectively than current strategies do and early implementation of appropriate treatments are both necessary.

## INTRODUCTION

Worldwide, hepatitis B virus (HBV) infection is a major cause of chronic hepatitis, liver cirrhosis, and hepatocellular carcinomas (*Beasley & Hwang, 1984*), and it continues to contribute to the most serious challenges currently posed by infectious diseases in public health. Although some countries, such as Taiwan, already have high immunization coverage, more than 257 million people worldwide have chronic HBV infection, with the majority of the infected people living in Africa and Asia (*WHO, 2017*). HBV and related complications result in nearly 600,000 deaths annually (*Perz et al., 2006*); therefore, despite the high vaccination coverage in many countries, HBV prevalence remains a major public health burden.

Prior to the introduction of the national HBV vaccination program in Taiwan in 1984, approximately 15%–20% of Taiwanese adults tested positive for the HBV surface antigen (HBsAg), with mother–infant vertical transmission being the primary means of infection (*Gust, 1996*; *Sung, 1984*). The nationwide HBV vaccination program was officially implemented in July 1984 in Taiwan. During the first two years of the program, the vaccination was available free-of-charge only to infants born to HBsAg-carrier mothers. A four-dose plasma-derived vaccine regimen was provided; the doses were administered at birth and at one, two, and 12 months of age (*Chen et al., 1987*). However, from July 1986 onwards, all infants were immunized against HBV by using the four-dose plasma-derived vaccine. Neonates born to highly infectious carrier mothers also received 0.5 mL of HBV immunoglobulin at birth. In addition, after November 1, 1992, the plasma-derived vaccine used for HBV vaccination was replaced by a recombinant yeast-derived vaccination with a three-dose regimen; the doses were administered at birth and at the ages of one and six months. From October 1990, the free catch-up HBV vaccination program was extended to include all children aged <7 years, all involved medical personnel, and selected groups of children (e.g., elementary-school children in aboriginal areas and offshore islands). The details of the program have been extensively documented previously (*Su et al., 2008*).

The program was highly successful, and within 12 years of its implementation, over 20 million vaccinations are estimated to have been provided to neonates, children, and secondary school and college students. Approximately 89% of the 3.2 million vaccinated infants completed their three-dose recombinant vaccination regimen, and at least 90% of the children currently aged <15 years have received the vaccination (*Ni et al., 2001*). Additional statistics from the Ministry of Health and Welfare, Taiwan, revealed that the 2016 newborn HBV vaccination rate was as high as 97.8%, which is the highest among the standard vaccines listed for infants (*CDC, 2017*). The program's effectiveness has been further demonstrated by numerous studies that have reported a decrease in vital HBV prevalence markers, namely overall HBsAg-carrier rate, chronic HBV infection incidence rate, and mother-to-child transmission infection rate, since the commencement of the vaccination program (*Ni et al., 2016*; *Ni et al., 2012*; *Ni et al., 2007*; *Su et al., 2007a*; *Su et al., 2007b*; *Tsen et al., 1991*).

Many studies have reported that neonatal HBV vaccination provides sufficient protection against HBV infections and that disease prevalence is significantly lower in the postvaccination generation than in the prevaccination generation (*Chien et al., 2006*; *Ni et al., 2016*; *Ni et al., 2012*; *Ni et al., 2007*; *Su et al., 2007a*; *Su et al., 2007b*). However, previous studies have primarily used unadjusted age for analysis and the cohorts and have rarely examined age, period, and cohort effects. According to our review of relevant studies, although HBV prevalence in Taiwan generally increases with age, this phenomenon becomes less apparent if the populations before and after vaccination are exclusively and independently observed, demonstrating that disease prevalence is likely to be affected by different time variables (*Chien et al., 2006*; *Ni et al., 2016*; *Ni et al., 2012*; *Ni et al., 2007*). Consequently, period and cohort effects should be considered when analyzing the relationship between age, vaccination, and disease.

Therefore, in this study, we examined whether age–period–cohort effects were significantly associated with HBV prevalence and determined whether disease prevalence was maintained below a threshold level after 30 years of universal neonatal HBV vaccination program implementation. Although the universal neonatal HBV vaccination program proved effective in both immunization coverage and disease prevention, continuous disease monitoring is crucial for disease control.

## MATERIAL AND METHODS

### Data collection and delinking

We collected data from the records of entrance health examinations of medical university students from 2005 to 2016 in northern Taiwan. The records contained data of undergraduate and graduate entrants. All data collected for this study were delinked by the university's office of environmental protection and occupational safety. However, demographic variables such as sex, the first letter of the social identification number, birth date, and serological markers of HBV, including seropositivity for HBsAg, HBV surface antibody (anti-HBs), and HBV core antibody (anti-HBc), were retained. Serum anti-HBc titer data were available only for 2012 and 2016. However, before delinking, the office screened and eliminated the duplicate records of undergraduates who completed their graduate studies in the same institution and retained only their initial undergraduate entrance records. Because the data collected for this study were delinked by the university's office of environmental protection and occupational safety, our study was approved, and the requirement of student informed consent was waived by the Joint Institutional Review Board of Taipei Medical University (N201603069).

### Seromarkers for HBV detection

The serum levels of HBsAg, anti-HBs, and anti-HBc were determined using a commercially available enzyme immunoassay kit (Elecsys 2010 system; Roche Diagnostics, Mannheim, Germany). The detection limit of the anti-HBs enzyme immunoassay kit was 0.1 mIU/mL. Samples with an anti-HBs titer of $\geq 10.0$ mIU/mL were interpreted as protective. Sample rate/cutoff rate (S/CO) for anti-HBc of $\leq 1.000$ were considered to represent reactivity to anti-HBc, and values between 1.001 and 3.000 were considered negative reactions. S/CO

of HBsAg ≥ 1.00 were considered to represent HBsAg reactivity, and samples with HBsAg < 1.00 were considered negative reactions.

## Definition of the age–period–cohort model

All participants were classified according to the stages of the national neonatal HBV vaccination programs. Cohort A consisted of students born before July 1984, before the implementation of the HBV vaccination program. Cohort B consisted of students born between July 1, 1984, and June 30, 1986, when the neonatal HBV vaccination was provided only to infants born to carrier mothers. Cohort C consisted of students born between July 1, 1986, and October 30, 1992, who received the plasma-derived HBV vaccine, and cohort D included students born after November 1, 1992 (who received the recombinant HBV vaccine). Cohorts C and D represented the period during which the vaccination program was extended to all newborns from July 1, 1986. Moreover, the students were grouped according to age. Periods were classified according to the entry year of the participants; accordingly, our model consisted of 12 periods from 2005 to 2016.

## Statistical analysis

The chi-squared ($\chi^2$) test was used for analyzing categorical variables, and descriptive data were presented as mean ± standard deviation. The SPSS 19.0 software package (Chicago, IL, USA) was used for data analysis, and $\alpha$ values <0.05 were considered statistically significant.

The age–period–cohort (APC) model was used to analyze the HBsAg status assuming that the model fit the Poisson regression distribution. A log-linear Poisson regression model was used for analysis:

$$\text{Formula: } \log\left(\frac{d_{ij}}{y_{ij}}\right) = \mu + \alpha_i + \beta_j + \gamma_k + \epsilon$$

where ($d_{ij}/y_{ij}$): the rate of interest,

$d_{ij}$: the number of cases in the $i$th age group and $j$th period,

$y_{ij}$: populations in the $i$th age group and $j$th period that are at risk,

$\alpha_i$: the effect of the $i$th age group,

$\beta_j$: the effect of the $j$th period, and

$\gamma_k$: the effect of the $k$th cohort.

The deviance and degrees of freedom were the factors used in the goodness-of-fit test. Variables were added to the age–period, age–cohort, and age–period–cohort models to analyze whether the results were altered; an alteration would imply that the factors affected the models. The deviance and degree of freedom were varied to determine whether the significance of the estimated $\chi^2$ value was changed for evaluation purpose.

# RESULTS

## Seroprevalence of HBV

Originally, a total of 18,783 students were selected for this study. Twenty students whose records lacked demographic data, 522 international students, and 630 students with incomplete data on HBV seromarkers were excluded. Finally, 17,611 students finally were

**Table 1  Hepatitis B virus infection status according to birth year of students who had enrolled between 2005 and 2016 at a university in northern Taiwan.**

| Birth year | Total | Sex | | HBsAg +ve | | | | HBsAg −ve, Anti-HBs +ve | | | |
|---|---|---|---|---|---|---|---|---|---|---|---|
| | | Female | Male | Total | Female | Male | P value[d] (for $\chi^2$) | Total | Female | Male | P value[d] (for $\chi^2$) |
| | n | n | n | n (%)[a] | n (%)[b] | n (%)[c] | | n (%)[a] | n (%)[b] | n (%)[c] | |
| −1974/6 | 1,250 | 841 | 409 | 121 (9.7) | 77 (9.2) | 44 (10.8) | 0.074 | 933 (74.6) | 654 (77.8) | 279 (68.2) | <0.001 |
| 1974/7–1979/6 | 958 | 699 | 259 | 98 (10.2) | 63 (9.0) | 35 (13.5) | <0.001 | 744 (77.7) | 559 (80.0) | 185 (71.4) | <0.001 |
| 1979/7–1984/6 | 1,633 | 1,131 | 502 | 108 (6.6) | 70 (6.2) | 38 (7.6) | 0.039 | 1,320 (80.8) | 945 (83.6) | 375 (74.7) | <0.001 |
| 1984/7–1986/6 | 969 | 619 | 350 | 41 (4.2) | 26 (4.2) | 15 (4.3) | 0.899 | 758 (78.2) | 505 (81.6) | 253 (72.3) | <0.001 |
| 1986/7–1988/6 | 2,132 | 1,309 | 823 | 39 (1.8) | 22 (1.7) | 17 (2.1) | 0.197 | 1,135 (53.2) | 737 (56.3) | 398 (48.4) | <0.001 |
| 1988/7–1990/6 | 2,366 | 1,428 | 938 | 26 (1.1) | 11 (0.8) | 15 (1.6) | <0.001 | 1,249 (52.8) | 802 (56.2) | 447 (47.7) | <0.001 |
| 1990/7–1992/6 | 2,317 | 1,388 | 929 | 31 (1.3) | 15 (1.1) | 16 (1.7) | 0.008 | 1,404 (60.6) | 884 (63.7) | 520 (56.0) | <0.001 |
| 1992/7–1994/6 | 2,267 | 1,373 | 894 | 16 (0.7) | 9 (0.7) | 7 (0.8) | 0.479 | 1,007 (44.4) | 626 (45.6) | 381 (42.6) | 0.004 |
| 1994/7–1996/6 | 1,980 | 1,188 | 792 | 6 (0.3) | 2 (0.2) | 4 (0.5) | 0.008 | 717 (36.2) | 440 (37.0) | 277 (35.0) | 0.058 |
| 1996/7–1999/6 | 1,739 | 1,062 | 677 | 10 (0.6) | 6 (0.6) | 4 (0.6) | 0.889 | 597 (34.3) | 370 (34.8) | 227 (33.5) | 0.258 |

Notes.
[a] Percentage of total population in the birth year.
[b] Percentage of total female population in the birth year.
[c] Percentage of total male population in the birth year.
[d] P value was used for comparing the percentages of female and male populations in a given birth year.
Abbreviations: HBsAg, HBV surface antigen; Anti-HBs, antibody against HBsAg.

included in the analysis. To analyze the seroprevalence of HBV, this study included the HBsAg and anti-HBs test results of students who had enrolled between 2005 and 2016. The chronic HBV infection (HBsAg positivity) rate decreased from 9.7% in university students born before June 1974 to <1% in students born after July 1992 (Table 1). The overall chronic HBV infection rate decreased considerably from 1979 to 1988 and has stayed below 1% since 1992 (Fig. 1). In our cohort, the rate of occurrence of chronic HBV carriers were higher in the male students than in the female students.

The proportions of HBV immunity due to hepatitis B vaccination or natural infection (HBsAg negative and anti-HBs positive cases) were calculated in each birth year group (Table 1). Two stages of decline were observed, first between the 1984–1986 (78.2%) and 1986–1988 (53.2%) cohorts and then between the 1990–1992 (60.6%) and 1992–1994 (44.4%) cohorts. The proportions were stable throughout the rest of the timeline, thus producing a stepwise pattern of change (Fig. 2). Female students tended to preserve higher HBV immunity (anti-HBs positivity) rates than their male counterparts did.

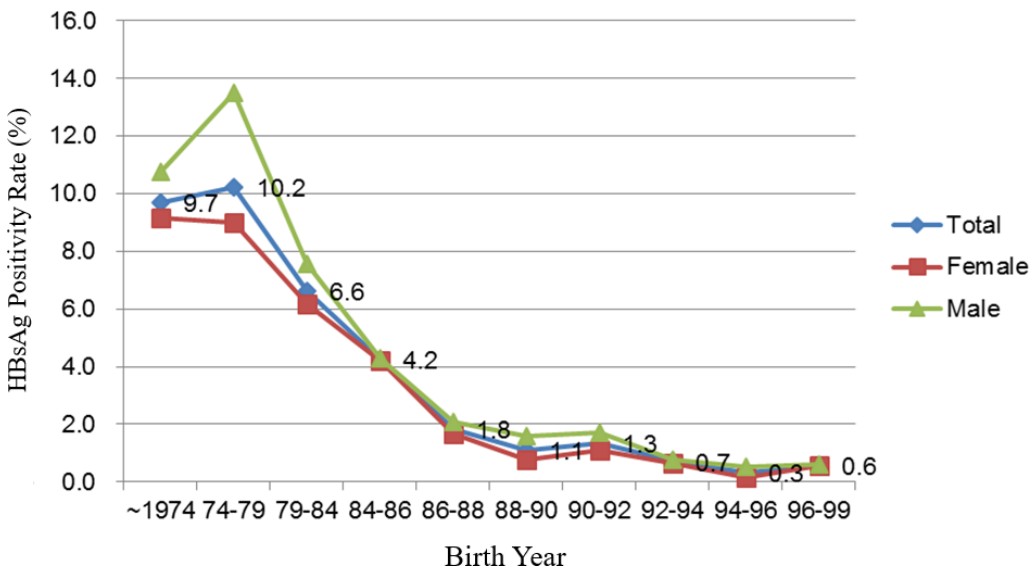

**Figure 1  Hepatitis B surface antigen positivity status of the students from a university in Taiwan who were enrolled between 2005 and 2016 according to birth year.**

According to our 2008 study, the prevalence of positive serum anti-HBc among the students in birth cohort years 1976, 1984, and 1986 were 48.7%, 9.8% and 4.0%, respectively. The prevalence of HBV immunity due to natural infection (HBsAg negative, anti-HBc positive, and anti-HBs positive) was 32.9%, 6.8%, and 2.1%, respectively (*Su et al., 2008*). The total prevalence rate of anti-HBc positivity in students aged 18–20 years was 4.3% in 2008. In this study, we also included the serum data of anti-HBc levels among students aged 18–20 years in 2012 and 2016, the mean values of which were 1.7% and 0.6%, respectively. The prevalence rates of HBV immunity due to natural infection were 0.5% and 0.6%, respectively, in 2012 and 2016.

## Age–period–cohort model

Table 2 presents the population distribution of the different birth cohorts. A total of 17,611 students were included in this study. A large proportion of the students belonged to cohort C, which constituted 41.0% of the study population. Cohort B had the fewest (968) students. The average age of the 17,611 students was 22.6 ± 6.8 years. Cohort A, with an average age of 32.8 ± 7.8 years, was significantly older than the other cohorts ($P < 0.001$) and had the largest age range: 21.1–70.5 years, whereas cohort D was the youngest, with an average age of 18.9 ± 1.0 years and the smallest age range: 16.8–23.7 years. Female students accounted for approximately 62.7% of the study population, but their distribution differed significantly among the cohorts ($P < 0.001$). If grouped according to their enrollment years, estimated 1,324–1,611 cases were tested each year. No case before 2010 was assigned to cohort D.

The APC model was developed to illustrate the relationship between temporal variables and chronic HBV infection (serum HBsAg positivity) among the student population

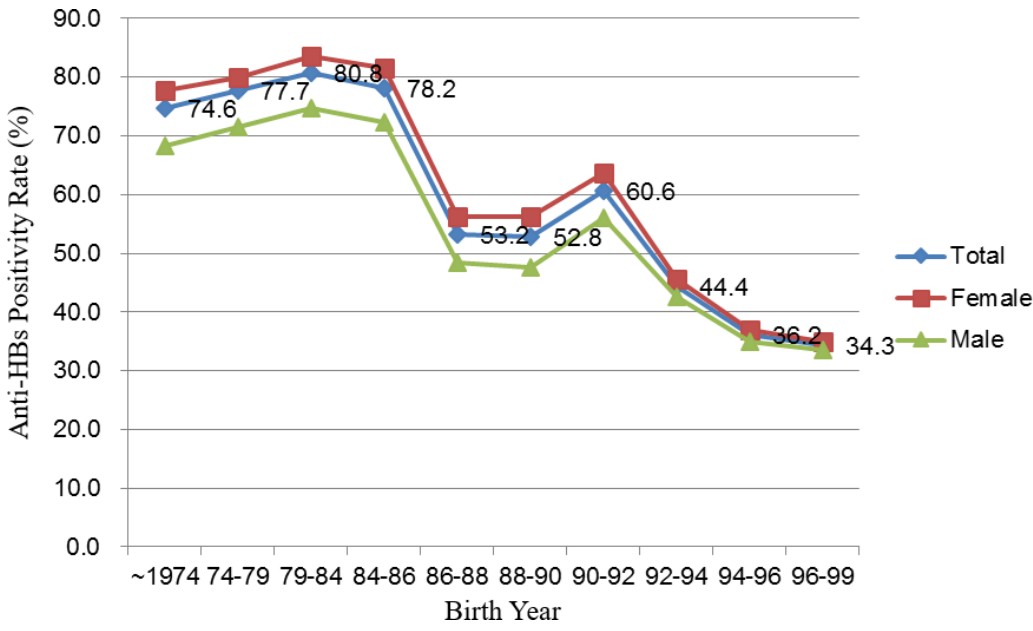

**Figure 2** Antibody against hepatitis B surface antigen positivity status of the students from a university in Taiwan who were enrolled between 2005 and 2016 according to birth year.

**Table 2** Distribution of students who had enrolled between 2005 and 2016 at a university in Taiwan into different age–period–cohort groups.

|  | Cohort A | Cohort B | Cohort C | Cohort D | Total |
|---|---|---|---|---|---|
| Sample size (%) | 3,842 (21.8%) | 968 (5.5%) | 7,220 (41.0%) | 5,581 (31.7%) | 17,611 |
| Age (mean ± SD) | 32.8 ± 7.8 | 23.4 ± 3.1 | 19.8 ± 2.4 | 18.9 ± 1.0 | 22.6 ± 6.8 |
| Female | 2,671 (69.5%) | 620 (64.0%) | 4,361 (60.4%) | 3,386 (60.7%) | 11,038 (62.7%) |
| Entry year |  |  |  |  |  |
| 2005 | 709 | 234 | 668 | 0 | 1,611 |
| 2006 | 651 | 67 | 830 | 0 | 1,548 |
| 2007 | 483 | 139 | 924 | 0 | 1,546 |
| 2008 | 326 | 163 | 950 | 0 | 1,439 |
| 2009 | 262 | 87 | 1,002 | 0 | 1,351 |
| 2010 | 350 | 48 | 1,036 | 6 | 1,440 |
| 2011 | 294 | 66 | 585 | 613 | 1,558 |
| 2012 | 199 | 47 | 368 | 871 | 1,485 |
| 2013 | 181 | 42 | 308 | 935 | 1,466 |
| 2014 | 127 | 30 | 209 | 958 | 1,324 |
| 2015 | 153 | 23 | 173 | 1,106 | 1,455 |
| 2016 | 107 | 22 | 167 | 1,092 | 1,388 |

**Notes.**

Cohort A: Students born before July 1, 1984;

Cohort B: Students born between July 1, 1984, and June 30, 1986, inclusively;

Cohort C: Students born between July 1, 1986, and October 30, 1992, inclusively;

Cohort D: Students born after November 1, 1992.

Abbreviation: SD, standard deviation.

**Table 3** Analysis of age–period–cohort effects of hepatitis B surface antigen positivity according to birth year in students who had enrolled between 2005 and 2016 from a university in Taiwan.

| Model | Degrees of freedom | Deviance | Likelihood ratio test | Effects | Deviance difference | df difference | P-value (for $X^2$) |
|---|---|---|---|---|---|---|---|
| Age | 15 | 620.3 | 31301.6 | | 492.7 | 9 | <0.001 |
| Period | 16 | 334.8 | 31755.2 | | 207.1 | 10 | 0.050 |
| Cohort | 12 | 415.4 | 31438.3 | | 287.7 | 6 | <0.001 |
| Age + period | 12 | 326.4 | 458.9 | Cohort | 198.7 | 6 | <0.001 |
| Age + cohort | 8 | 347.3 | 275.5 | Period | 219.6 | 2 | <0.001 |
| Period + cohort | 9 | 320.3 | 323.6 | Age | 192.6 | 3 | <0.001 |
| Age + period + cohort | 6 | 127.7 | 255.8 | | Reference | Reference | |

**Notes.**

Abbreviation: *df*, degrees of freedom.

enrolled between 2005 and 2016 in a medical university in northern Taiwan. Table 3 shows the deviance and likelihood ratio test results of age, period, cohort, and their combination effects on the APC models. Among these models, the age, period, and cohort effects were significant at the 5% level (all $P < 0.001$). However, magnitude of the age effect was the highest (deviance difference = 492.7, $df = 9$) when the age + period + cohort effect was the reference. The cohort effect was also highly significant (deviance difference = 287.7, $df = 6$). Therefore, the magnitude of the age + cohort effect was the highest (deviance difference = 219.6, $df = 2$) among the age–period, period–cohort, and age–cohort models.

## DISCUSSION

The results of this study suggested that since the implementation of the national neonatal HBV vaccination program, the overall chronic HBV carrier rate has significantly decreased and has consistently remained <1% in the student population after the birth year 1992. In addition, a significant decrease was observed in the anti-HBs prevalence rates in the cohorts after the implementation of the vaccination program. An additional decline in the prevalence of anti-HBs positivity was observed when the national plasma-derived vaccine scheme was replaced by the recombinant vaccine scheme in 1992. The national HBV vaccination program was a major strategy implemented for the prevention of HBV infection over the years. Our study, which is one of the first to use an APC model to assess the efficacy of the national HBV vaccination program, suggested that the program has been effectively preventing HBV infection since its commencement.

In our study, after the implementation of the national HBV vaccination program in 1984, the prevalence rate of chronic HBV infection declined from 4.2% (males: 4.3% vs females: 4.2; $P = 0.899$) to approximately 0.6% (males: 0.6% vs females: 0.6%; $P = 0.889$) in the university students born in 1999. Furthermore, according to our 2008 study, the prevalence rates of positive serum anti-HBc in the students in birth cohort years 1976, 1984, and 1986, were 48.7%, 9.8%, and 4.0%, respectively, and the total prevalence rate of anti-HBc positivity among the students aged 18–20 years was 4.3% (*Su et al., 2008*). Because the presence of anti-HBc indicates previous or ongoing infection with HBV, in

this study, we also included the serum anti-HBc titer data of students aged 18–20 years in 2012 and 2016, the mean values of which were 1.7% and 0.6%, respectively. These observations suggest that the implementation of HBV vaccination program reduced the prevalence of natural HBV infection, which is consistent with the findings of previous studies. A 2017 Taiwanese study demonstrated that the prevalence rates of chronic HBV infection in the plasma-derived and recombinant vaccination groups were 1.5% and 0.3%, respectively, in 38,000 university students recruited from 2003 to 2015 (*Hsu et al., 2017*). In addition, *Ni et al. (2012)* reported that 25 years after the introduction of the universal neonatal HBV vaccination program in Taiwan, the seroprevalence of HBV decreased from 9.8% (prevaccination period) to <1%. *Ni et al. (2016)* also noted that among 3,299 HBV-vaccinated young people, only 0.5% tested positive for HBV infection 30 years after the national neonatal HBV vaccination program. This suggested that among the younger students who were born after the implementation of the HBV vaccination program in 1984, the prevalence rate of chronic HBV infection decreased to <1%. Taiwan has already transformed from a hyper- to a low-endemic region for HBV infection (*Ni et al., 2016*).

Results from our previous studies and the current study have also suggested a strong association between changes in the policies of the national HBV vaccination program and the HBV immunity status of our students in different birth cohort years. The prevalence of chronic HBV infection decreased significantly after the implementation of the national HBV vaccination program in July 1984. We also observed a decline in the anti-HBs positivity among the birth year groups of July 1984 to June 1986, July 1986 to June 1992, and after July 1992, which represent the periods in which the plasma-derived HBV vaccination was administered to high-risk infants, plasma-derived vaccination was administered to all infants, and recombinant vaccination was administered to all infants, respectively. The stepwise decreasing trend in HBV immunity (anti-HBs positivity) shown in Fig. 2 can also be explained using the time frame of national HBV vaccination policy change. The persistence of HBV immunity (anti-HBs positivity) in students born before 1984 is primarily due to the high percentage of HBV immunity caused by natural infection. With the implementation of the HBV vaccination program in July 1984, passive immunity due to HBV vaccination (HBsAg negative, anti-HBs positive, and anti-HBc negative) has become the predominant constitution of the HBV immunity. Some previous studies have reported higher disappearance rates of anti-HBs after recombinant HBV vaccination than after plasma-derived vaccination within 12–15 years after primary vaccination (*Floreani et al., 2004*; *Kao et al., 2009*). Consequently, the prevalence of HBV immunity among young students born after 1992 decreased sharply when the recombinant HBV vaccine was introduced in November 1992. Moreover, in our study, the 1992–1994 birth year was defined from July 1, 1992, to June 30, 1994. The administration of the recombinant three-dose yeast-derived vaccine commenced from November 1, 1992. Hence, the students who were born between July 1, 1992, and October 30, 1992, received the four-dose plasma-derived vaccine; this may satisfactorily explain why the mean anti-HBs antibody level in the 1992–1994 group (which received plasma-derived and recombinant HBV vaccines) was higher than the mean levels in the 1994–1996 and 1996–1999 groups (which received only recombinant HBV vaccine).

Hence, our observation suggests that the prevalence rates of HBV immunity (anti-HBs positivity) in fact continuously declined from the group that acquired it from anti-HBs natural-boosting and the group that acquired it from plasma-derived HBV vaccines, to the group that acquired anti-HBs positivity from the recombinant vaccine. By contrast, *Yuen et al. (1999)* reported a nonsignificant difference between plasma-derived and recombinant vaccines in children aged <11 years. However, in a recent study, Li et al. reported that students in the plasma-derived group exhibited a higher persistence rate of anti-HBs positivity (43.2%) than did their recombinant counterparts (33.3%) *Li et al. (2015)*. They also suggested that the difference in antigen content between the neonatal plasma-derived and recombinant vaccines might have influenced the antigen persistence and immune memory (*Lin et al., 2011*). *Samandari et al. (2007)* conducted a study in Alaska, which reported that 11- to 14-year-old adolescents who had received the plasma-derived vaccine at birth had 7% higher rates of anti-HBs positivity than their counterparts who had received the recombinant vaccine (14%) in the same age group. *Kao et al. (2009)* also observed that among adolescents aged 13–15 years, anti-HBs positivity was exhibited by 64.5% of the group that had received the plasma-derived vaccination (plasma-derived group) born to mothers with chronic HBV infection between July 1984 and June 1986, 44.1% of the plasma-derived group born between July 1986 and June 1992, and 36.0% of the recombinant group born between July 1992 and June 1995 ($P < .001$). In a more recent study, *Hsu et al. (2017)* reported that the seroprevalence of HBV immunity in the recombinant group was 39% lower than that in the plasma-derived group. However, both types of vaccines have been reported to be comparable in terms of long-term immunogenicity and protective efficacy in many studies conducted in different populations (*Dentinger et al., 2005*; *Kao et al., 2009*). The persistence of serum anti-HBs level is likely affected by the types of vaccines, brands, doses, and times of administration of the primary HBV vaccination as well as by high re-exposure rates to HBV (*Hsu et al., 2017*; *Yuen et al., 1999*).

In this study, sex disparity was observed in chronic HBV infection, particularly in the prenational HBV vaccination era, and in the reduced immune response to the HBV vaccine. Several factors have been reported to adversely affect the antibody response to HBsAg including the site and route of injection, sex, advanced age, body mass (being overweight), nutritional status, smoking, and genetic factors (*Chang, 2006*; *Morris et al., 1989*; *Zuckerman, 2006*). In one previous Taiwanese study, chronic HBV carrier prevalence rate was also observed to be higher in male than in female students (10.7% vs 4.4%) who born prior to the introduction of national neonatal HBV vaccination and were followed up for >18 years in Taiwan (*Su et al., 2007a*). In another Taiwanese case–control study, after adjusting for confounding factors, the elevated baseline of serum HBV titer was found to be significantly associated with male HBV carriers (*Chen et al., 2009*).

By setting HBsAg as the random variable for constructing the APC model, we could investigate any significant difference in the prevalence under different variables of time. The results revealed that the prevalence rates of chronic HBV infection in our student population exhibited significant age, period, and cohort effects ($P < 0.001$). A recent population-based survey and longitudinal follow-up study conducted in Taiwan demonstrated that the seromarkers of HBV infection generally increase with age in the pre- and postvaccination

generations (*Chen et al., 2015*). Multiple studies have discussed the period and cohort effects in detail and have reported a significant decrease in HBV seromarkers in the cohorts after the implementation of the vaccination program (*Ni et al., 2016*; *Ni et al., 2012*; *Ni et al., 2007*). Small-scale clinical trials have also consistently established that neonatal vaccinations can effectively prevent acute and chronic HBV infection (*Bruce et al., 2016*; *Dumaidi & Al-Jawabreh, 2015*; *Huang et al., 2015*; *Su et al., 2013*). The significant cohort effect further confirms that HBV disease prevalence decreases with each generation. Previous seroepidemiological studies in Taiwan have demonstrated an HBV vaccination coverage rate ($\geq$3 doses of vaccine) of 89.5% in Taiwan in 1986, 92.8% before December 2002, and 97% in the 2002 birth cohort (*Gust, 1996*; *Lin et al., 2011*; *Ni et al., 2007*). The vaccination coverage rate has already reached its peak plateau and is likely among the highest globally. The implementation of the national HBV vaccination program in 1984 was a major intervention tool for combating HBV infection, and our APC model demonstrated that the program has been highly successful in preventing HBV infection in young adults.

With the implementation of national HBV vaccination and the accompanying measures in public health and preventive medicine, the majority of HBV acquisition in the post-HBV vaccination era has been successfully controlled. A recent study demonstrated that approximately 80% of the HBV carriers born in the post-HBV vaccination era had a positive maternal status, and the authors suggest that mother–infant vertical transmission remains a crucial cause of vaccine failure (*Ni et al., 2016*). Additional prenatal precautionary measures should be considered in addition to the routine implementation of the HBV vaccination program for total HBV eradication.

Some limitations of this study must be discussed. Because this data set was delinked, the vaccination history of the participants and the carrier status of the participants' mothers were not available. However, our HBV carrier and anti-HBs positivity rates were comparable with those reported in previous studies. Our HBV vaccine coverage rate can be assumed to be similar to that of other studies in Taiwan because all the university new entrants were under the same national HBV scheme. Second, the data set was collected from a single university in northern Taiwan. However, the university admission was based on national university and graduate school entrance examination results. Consequently, the students were from various parts of Taiwan, which minimized the selection bias. Third, a slight chance of duplicate data may exist because some students may possess health records at both college and graduate levels in the same school. However, this selection bias was minimized through a careful cross verification prior to the identification delinking process. Finally, we did not provide the annual anti-HBc data representing the status of HBV natural infection. However, by using the data of our previous study in 2008 and the data of anti-HBc status in 2012 and 2016, we presented a general declining trend in anti-HBc positivity in our study.

## CONCLUSION

This study found that the prevalence rate of chronic HBV infection decreased to <1% 30 years after the implementation of national HBV vaccination in 1984. Our APC model

revealed that HBV vaccination is one of the most effective strategies for preventing HBV infection. The HBV vaccination program has transformed Taiwan from a hyper- to a low-endemic region. However, the complete eradication of HBV infection requires additional effective strategies for detecting vaccination failure and the early implementation of appropriate treatments. Further research is warranted for investigating these strategies.

### Funding

This study was funded by the Ministry of Science and Technology, Taiwan (105-2314-B-038-038) and the Taipei Medical Hospital (105TMU-TMUH-19). The funders had no role in study design, data collection and analysis, decision to publish, or preparation of the manuscript.

### Grant Disclosures

The following grant information was disclosed by the authors:
Ministry of Science and Technology, Taiwan: 105-2314-B-038-038.
Taipei Medical Hospital: 105TMU-TMUH-19.

### Competing Interests

The authors declare there are no competing interests.

### Author Contributions

- Yang-Cheng Hu performed the experiments, analyzed the data, wrote the paper, prepared figures and/or tables, reviewed drafts of the paper.
- Chih-Ching Yeh conceived and designed the experiments, performed the experiments, analyzed the data, contributed reagents/materials/analysis tools, prepared figures and/or tables, reviewed drafts of the paper.
- Ruey-Yu Chen conceived and designed the experiments, performed the experiments, contributed reagents/materials/analysis tools, reviewed drafts of the paper.
- Chien-Tien Su performed the experiments, contributed reagents/materials/analysis tools, reviewed drafts of the paper.
- Wen-Chang Wang analyzed the data, contributed reagents/materials/analysis tools, reviewed drafts of the paper.
- Chyi-Huey Bai analyzed the data, contributed reagents/materials/analysis tools, prepared figures and/or tables, reviewed drafts of the paper.
- Chi-Fei Chan wrote the paper, reviewed drafts of the paper.
- Fu Hsiung Su conceived and designed the experiments, performed the experiments, analyzed the data, contributed reagents/materials/analysis tools, wrote the paper, prepared figures and/or tables, reviewed drafts of the paper.

### Human Ethics

The following information was supplied relating to ethical approvals (i.e., approving body and any reference numbers):

As the data collected for this study were delinked by the university's office of environmental protection and occupational safety, we have been waived of patient informed consent by the Joint Institutional Review Board of Taipei Medical University (N201603069).

## Data Availability

The raw data has been provided as Data S1.

## Supplemental Information

Supplemental information for this article can be found online at http://dx.doi.org/10.7717/peerj.4297#supplemental-information.

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
