# Peer review of "Seroprevalence of hepatitis B virus in Taiwan 30 years after the commencement of the national vaccination program"

_PeerJ, doi:10.7717/peerj.4297_

## Round 0.1 · original submission · Minor Revisions

Please read the reviewers’ comments carefully and respond to them. Furthermore, the manuscript needs some English language editing, before resubmission.

·

Basic reporting

In introduction section, author shew context of hepatitis B infection and prevalence globally. There are some data need to be updated.
1. Author mentioned that hepatitis B virus infection is a major cause of acute hepatitis, this is not precise because most of the acute infections become immune and the major problem is chronic infection, since majority of chronic infections are lifelong.
2. Author cited literature and said “More than 350 million people worldwide remain chronically infected with HBV”, which should be updated by the WHO new published document (Hepatitis Report 2017, WHO).

Experimental design

Suggest to make close linkage between birth cohort and policies change

Validity of the findings

1. What makes difference of HBsAg prevalence is not the birth cohort, but the policies change. Therefore, author should have more discussions about the policies rather than the birth cohort.
2. The HBsAg positivity rate decreased from 9.7% among patients born before June 1994 to <1.0% among patients born after 1992. There is a cross time, please ckeck.
3. See Table 2. Author had clear criteria for age of each group, why there is an overlap for age range. Please clarify.
4. Table 1 needs to be restructured in a more standard way.
5. Figure 2, Can author provide more information and explanation why there is a stepwise decline for anti-HBs?

Additional comments

This manuscript is written in simple and common English language, need more professional.

Reviewer 2 ·

Basic reporting

-

Experimental design

-

Validity of the findings

-

Additional comments

Hu et al. report seroprevalence of hepatitis B virus (HBV) infection in Taiwan, after 30 years of the initiation of the national HBV vaccination program, since 1984. To explore the long-term efficacy of the HBV vaccination program, the investigators analyzed available data on the presence of HBV surface antigen (HBs Ag) and anti-HBs antibody in the sera of a total of 17,611 students enrolled into a Taiwanese university, during 2005-2016. I have some comments that might help to improve the manuscript, below:

-The title and other places in the manuscript: the term 'hepatitis B infection' should be clearly stated as 'hepatitis B virus infection'.

-The term 'patients' used in the abstract and the main text may not be accurately defined, since the subjects here are students with no diseases.

-Line 67: duplicate words (that that).

-Line 67-68: four- or three-dose plasma-derived vaccination? please check the number of time points for vaccination, mentioned in the sentence.

-Line 74, 77, 78, and other places: "hepatitis B (HB)" should be clearly stated as 'hepatitis B virus (HBV)'.

-Line 99-111: the Methods '2.1. National HBV vaccination schedule' - This section seems to be more appropriate to be incorporated in, as a part of, the introduction.

-Line 130: titer or level?

-Line 130: Please provide a reference for the criteria "anti-HBs Ag antibody < 10 mIU/mL is non-protective". Is it possible to have the immuno-protecitve state against HBV with the detectable anti-HBs Ab level < 10 mIU/mL?

-Line 131-132: Please check the criteria for setting 'reactive' VS 'non-reactive' for anti-HBc antibody. Should 'S/CO > 1' be considered as reactive (not non-reactive nor just 1-3)?

-Should not begins a sentence with number, i.e., Line 168 and 170.

-Line 196: please consider to delete 'due to sampling restriction'.

-Line 198-203: This paragraph, as it currently stands, is more like discussion. The APC model analysis seems to be the key information of this manuscript, so please revise this section to report clearly the facts/results (with some details). This includes brief descriptions of some key parameters used in Table 3 (perhaps as footnote), so that the readers who are not familiar with this kind of APC model analysis can follow.

-Line 175-176 mentioned the HBV carrier state in males > females, and in contrast, Line 181-182 mentioned the anti-HBsAg antibody level in male < female. Since the male-to-female ratio < 1 (i.e., as shown in Table 1), what is/are reason(s) behind this paradoxical observation, in regard of gender and HBV carriers?

-Line 183-186: Anti-HBc antibody can be used to discriminate the HBV protective immunity from natural infection and vaccination. The anti-HBc antibody dropped from 4.3% (in 2008) to 1.7% (in 2012) and 0.6% (in 2016). What does it mean? Rate of natural HBV infection is decreased? What about the other markers (i.e., HBsAg and anti-HBsAg) in this groups of students?

-The discussion could be shortened, by for example, removing repeated statements (i.e., the first sentence of the first paragraph), and perhaps shortening/removing the statements that are considerably non-relevant to the results of the current study (i.e., Line 284-296).

-Regarding the persistence of anti-HBs antibody, Cohort C (students were born during 1986-1992, and received the plasma-derived vaccine), the mean anti-HBs antibody level was observed to be higher in the 1990-92 birth year group, compared to that of the 1986-88 and 1988-90 groups. This observation can be expected, because the 1989-92 group received the vaccine in later time point than the other 2 groups. However, for Cohort D (students were born during 1992-99, and received the recombinant vaccine), I wonder why the mean anti-HBs antibody level of the 1996-99 group was observed to be lower than the students born and received the vaccine earlier (i.e., 1994-96 and 1992-94).

---

## Round 0.2 · accepted · Accept

Please address the comment of reviewer 1 while in production.

·

Basic reporting

All fit, no comment.

Experimental design

No additional comment.

Validity of the findings

No additional comment.

Additional comments

Author responsed: We have updated this text accordingly in lines 66–67 on page 3: “more than 350 million people worldwide have chronic HBV infection, with the majority of the infected people living in Africa and Asia (WHO 2017).” However,
I checked The Global Hepatitis Report 2017(WHO), it says that:
WHO estimates that in 2015, 257 million persons, or 3.5% of the population, were living with chronic HBV
infection in the world.
Hence, 257 million should uodate 350 million.

Reviewer 2 ·

Basic reporting

-

Experimental design

-

Validity of the findings

-